# Predicting partner fitness based on spatial structuring in a light-driven microbial community

Jonathan K. Sakkos[1], María Santos-Merino[1], Emmanuel J. Kokarakis[1,2], Bowen Li[3], Miguel Fuentes-Cabrera[4], Paolo Zuliani[5], Daniel C. Ducat[1,6]*

**1** Plant Research Laboratory, Michigan State University, East Lansing, Michigan, United States of America, **2** Department of Microbiology & Molecular Genetics, Michigan State University, East Lansing, Michigan, United States of America, **3** School of Computing, Newcastle University, Newcastle upon Tyne, United Kingdom, **4** Center for Nanophase Materials Sciences, Oak Ridge National Laboratory, Oak Ridge, Tennessee, United States of America, **5** Dipartimento di Informatica, Università di Roma "La Sapienza", Rome, Italy, **6** Department of Biochemistry and Molecular Biology, Michigan State University, East Lansing, Michigan, United States of America

* ducatdan@msu.edu

**Data Availability Statement:** All relevant data are within the manuscript and its Supporting Information files. The modified NUFEB source code and examples are publicly available at: https://

## Abstract

Microbial communities have vital roles in systems essential to human health and agriculture, such as gut and soil microbiomes, and there is growing interest in engineering designer consortia for applications in biotechnology (*e.g.*, personalized probiotics, bioproduction of high-value products, biosensing). The capacity to monitor and model metabolite exchange in dynamic microbial consortia can provide foundational information important to understand the community level behaviors that emerge, a requirement for building novel consortia. Where experimental approaches for monitoring metabolic exchange are technologically challenging, computational tools can enable greater access to the fate of both chemicals and microbes within a consortium. In this study, we developed an *in-silico* model of a synthetic microbial consortia of sucrose-secreting *Synechococcus elongatus* PCC 7942 and *Escherichia coli* W. Our model was built on the NUFEB framework for Individual-based Modeling (IbM) and optimized for biological accuracy using experimental data. We showed that the relative level of sucrose secretion regulates not only the steady-state support for heterotrophic biomass, but also the temporal dynamics of consortia growth. In order to determine the importance of spatial organization within the consortium, we fit a regression model to spatial data and used it to accurately predict colony fitness. We found that some of the critical parameters for fitness prediction were inter-colony distance, initial biomass, induction level, and distance from the center of the simulation volume. We anticipate that the synergy between experimental and computational approaches will improve our ability to design consortia with novel function.

## Author summary

Microbial communities, play important, yet poorly understood roles in health and agriculture. As we develop a better understanding of how these communities interact

github.com/jsakkos/NUFEB and https://github.com/nufeb/NUFEB. Parameter optimization and data analysis notebooks are available here: https://github.com/Jsakkos/nufeb-cyano-e-coli.

**Funding:** This work was funded by Department of Energy Grant DE−FG02−91ER20021 (to D.C.D.) at the MSU DOE-PRL. Additional support for the research was provided by NSF Award 845463 (to D.C.D.). This research used resources of the Compute and Data Environment for Science (CADES) at the Oak Ridge National Laboratory, which is supported by the Office of Science of the U.S. Department of Energy under Contract No. DE-AC05-00OR22725. The funders had no role in study design, data collection and analysis, decision to publish, or preparation of the manuscript.

**Competing interests:** The authors have declared that no competing interests exist.

together, as well as with their host organisms, there is growing interest in engineering communities with specific functions, such as for treating disease, personalized probiotics, or aiding plants with nutrient uptake. To better understand how these microbes interact with each other, we want to monitor the exchange of metabolites and the locations of the microbes, tasks which at present are technically challenging, if not impossible. Where experimental approaches for monitoring metabolites are limited, computational tools can enable greater access to the fate of both chemicals and microbes within a community. In this study, we developed a computerized model of a synthetic microbial community of two bacteria, one which performs photosynthesis and supplies sugar and another which consumes the sugar for growth. We showed that the relative level of sugar secretion regulates not only the steady-state support for the consumer partner's growth, but also how the community changes with time. To determine the importance of spatial organization within the community, we fit a model and used it to predict colony growth. We anticipate that the synergy between experimental and computational approaches will improve our ability to design microbial communities with new functions.

## Introduction

With recent advances in molecular tools and the ubiquity of genetic components, mixed species cultures composed of engineered microbes are attracting increased interest. Engineered communities could potentially advance a wide range of biotechnological applications from human health to agriculture [1–4]. Synthetic microbial consortia consist of two or more tractable cell populations, which typically have well-developed genetic toolkits (*e.g.*, synthetic promoters, ribosome binding sites, reporter proteins, etc.). Modular genetic components can be assembled into functional circuits and metabolic pathways which control cellular [5–8] as well consortia function [9–12].

To date, the scope and complexity of consortia that have been constructed has been limited. Most studies consist of pairs of auxotrophic mutants of the same species that cross-exchange essential amino acids and/or utilize a common carbon source [7,11,13,14]. Additionally, synthetic consortia are typically fragile, with some requiring physical separation to prevent the extinction of partners [15], vulnerable to environmental perturbations, and susceptible to invasive species. There is a desire to construct consortia which are robust and can be used in heterogeneous environments (*e.g.*, bioreactors, outdoor sloughs) for real-world applications. However, the impact of local micro-environments and nutrient concentration gradients represent an often-under-explored feature that strongly impacts performance within synthetic communities.

In nature, microbes routinely produce and secrete chemical resources which benefit other cells, often referred to as a "public good." Despite the metabolic burden of producing public goods, cooperating partners can mutually benefit from their respective exchanges; a division of labor emerges based on specialization [16]. This cooperativity makes the population more robust to a variety of environmental challenges [17–21]. In natural symbiotic interactions, spatial organization plays an important role in promoting stable interactions by facilitating rapid exchange of metabolites and promoting repeated interactions between species [22]. For example, biofilms (extracellular matrix-encased accretions which adhere to surfaces) define spatial organization in consortia, form chemical gradients as a result of limited diffusion and heterogeneous intercellular spacing, and create micro-environments that regulate the behavior of individual cells [17,20]. Unfortunately, even in the best-studied natural consortia, it is

technically challenging to evaluate, monitor, and control the flux of metabolite exchange. Complementing experimental research with computational approaches targeting well-defined species can provide valuable insights into the underlying phenomena.

Individual-based Models (IbM) are a bottom-up approach that can simulate how the heterogeneity of individuals and local interactions influence emergent behaviors within a population. When applied to microbial consortia, individuals often refer to microbes, each with their own set of attributes (*e.g.*, growth rate, position, size). Modes of interaction may include cooperation, parasitism, commensalism, or competition resulting from complex processes such as cell secretion, division, utilization, and metabolization of nutrients. Emergent behaviors that we are interested in include species distribution, population dynamics, etc. Unlike with experimental observation, the state within an IbM is fully defined, allowing for exploration of the complex and nuanced relationships within synthetic consortia.

We have developed a cyanobacterial platform for constructing synthetic microbial consortia which is modular, stable, and flexible. The core cyanobacterial species of this platform is *Synechococcus elongatus* PCC 7942 engineered to express sucrose permease (*cscB*) and a heterologous copy of sucrose phosphate synthase (*sps*). We have previously characterized strains of *S. elongatus* bearing inducible copies of *cscB* and *sps*, (hereafter referred to as *S. elongatus* CscB/SPS), and have shown that they can secrete large quantities of sucrose (up to ~80% of total fixed carbon), a readily metabolized feedstock [23–26]. These cyanobacterial strains have been characterized extensively by our lab and others [23,25–30], and have been shown to directly support co-cultivated heterotrophic microbes in communities that are stable over long time periods [30–34]. Unlike many other synthetic consortia, the strength of cooperation with this platform can be controlled directly, allowing for experimentally tuning the extent of interspecies interaction [23,25,26,31]. When engineering consortia with this platform, we wish to understand the primary factors impacting partner fitness, and the complementary nature of IbMs were well-suited to this task.

In this study, we utilized experimental data from axenic phototrophic and heterotrophic cultures to construct, optimize, and validate an IbM implemented using the NUFEB framework (Fig 1) [35]. We used *Escherichia coli* W (hereafter *E. coli*) as the heterotrophic partner in our synthetic consortia, which has been extensively characterized [36–38] and which has been

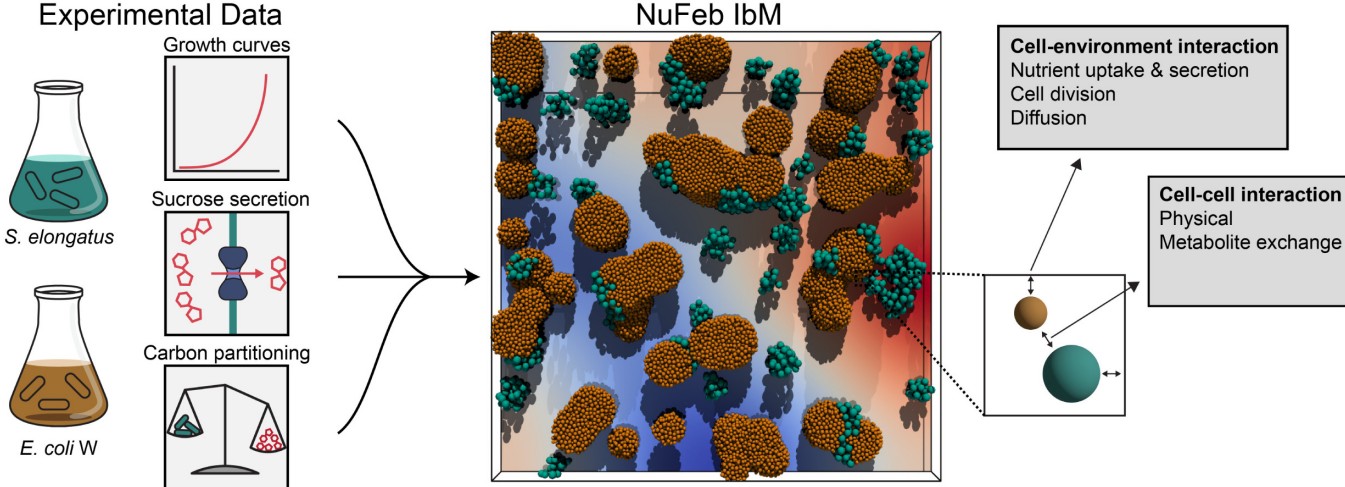

**Fig 1. Study overview diagram.** Experimental growth, sucrose secretion, and biomass apportionment data were used to construct an Individual-based NUFEB model of an *S. elongatus* and *E. coli* consortium.

previously shown to be capable of forming a stable consortium with sucrose-secreting *S. elongatus* [31,39]. Unlike some other industrially relevant strains, *E. coli* W can efficiently utilize sucrose as its sole source of carbon, which is critical for the construction of synthetic consortia with *S. elongatus* CscB/SPS. NUFEB simulations were used to understand how sucrose secretion levels affect not only the relative population levels, but also the dynamics between partner microbes. We reaffirm the importance of spatial organization and developed a model to predict partner fitness based on the analysis of colony-level metrics.

## Materials & Methods

### Microbial culturing conditions

*S. elongatus* CscB/SPS [25] cultures (50–100 mL) were grown in baffled flasks (Corning) with BG-11 medium (Sigma) supplemented with 1 g L$^{-1}$ HEPES, pH 8.3, in a Multitron II shaking incubator (Infors HT). Cultures were grown under continuous light with GroLux bulbs (Sylvania) at 125 μmol photons m$^{-2}$ s$^{-1}$, 2% $CO_2$, 32˚C. Cultures were back-diluted daily to an $OD_{750}$ of 0.3 and acclimated to the medium/irradiance for at least 3 days prior to isopropyl-β-D-thiogalactoside (IPTG) induction. Where appropriate, 1 mM IPTG was added to induce *cscB* and *sps* gene expression.

 *E. coli* cultures were grown in 250 mL baffled flasks using R2 medium [36,40] supplemented with up to 20 g/L sucrose @ 32˚C, with shaking at 260 rpm.

### Sucrose quantification

Secreted sucrose was quantified from supernatants using the Sucrose/D-Glucose Assay Kit (K-SUCGL; Megazyme) and converted into cell basis via $OD_{750}$ standard curve calibration.

### Dry cell weight measurement

Dry cell mass was determined as previously described [26]. *S. elongatus* cultures (~47.5 mL) were harvested after 24 hr post-induction by centrifugation at 4,000 rpm for 30 min. Pellets were washed twice with distilled water and transferred onto cellulose acetate membranes (0.45 μm, Whatman) and immediately dried in a hot air oven for ≥4 hr at 90˚C. The mass of each membrane was measured with an analytical balance before and after adding the cells, and these data were used to calculate the dry cell weight per volume.

### Individual-based model

Individual-based representations of sucrose secreting *S. elongatus* CscB-SPS and *E. coli* was developed and integrated into NUFEB (Newcastle University Frontiers in Engineering Biology) [35]. In the IbM, the computational domain is defined as the volume where bacterial cells reside and where the biological, physical, and chemical processes take place. Within this area, nutrients are represented as a continuous field where their dynamics over time and space are updated at each discrete Cartesian grid. Cells are modelled as spherical particles with each having a set of state variables to describe its physical and biological attributes (position, size, growth rate, etc.). These attributes vary between cells and can change over time because of external or internal processes. The processes that influence cellular activities are classified into three sub-models: biological, physical, and chemical. The biological sub-model handles metabolism, growth, and reproduction. An individual cell grows and its mass increases by consuming nutrients in the grid where it is located. The equation governing the mass $m_i$ of cell $i$ is

**Table 1. Bacterial physical parameters.**

| Species | Parameter | Value | Unit | Reference |
|---|---|---|---|---|
| *S. elongatus* | Length | | μm | [44] |
| | Min | 2.5 | | |
| | Max | 5 | | |
| | Diameter | 1 | μm | [44] |
| | Volume-equivalent diameter | | μm | |
| | Min | 1.37 | | |
| | Max | 1.94 | | |
| | Density | 370 | g/L | Calculated from [44] |
| *E. coli* | Length | | μm | [44] |
| | Min | 1.6 | | |
| | Max | 4 | | |
| | Diameter | 0.8 | μm | [44] |
| | Volume-equivalent diameter | | μm | |
| | Min | 0.88 | | |
| | Max | 1.39 | | |
| | Density | 230 | g/L | Calculated from [45] |

given by:

$$\frac{dm_i}{dt} = \mu_i m_i, \tag{1}$$

where a Monod-based growth model is implemented to determine the specified growth rate μ. Species-specific models were implemented based on both experimental data collected in this work and prior studies. See Tables 1 and 2 for physical and growth parameters, respectively. The cyanobacteria model depends solely on (the concentrations of) light, $CO_2$, and IPTG:

$$\mu_i = \mu_{max}\left(\frac{[light]}{K_{light} + [light]}\right)\left(\frac{[CO_2]}{K_{CO_2} + [CO_2]}\right)\left(0.141 e^{\frac{-[IPTG]}{0.063}} + 0.9\right) \tag{2}$$

$$\psi_i = \mu_i\left(-3.4897 e^{\frac{-[IPTG]}{0.048}} + 3.4092\right) \tag{3}$$

where $\mu_{max}$ is the maximum growth rate coefficient (s$^{-1}$), $K_{light}$ /$K_{CO2}$ are the half-velocity constants for light and $CO_2$, respectively, and the right-hand portion of Eq 2 is an empirical fit for growth reduction with respect to IPTG induction of the sucrose secretion machinery (CscB/SPS), and Ψ is the metabolic flux due to sucrose secretion. Nutrient mass balance in the system is defined by the following equations:

$$\frac{\partial S}{\partial t} = \nabla \cdot (D\nabla S) + R, \tag{4}$$

where ∇ is the gradient operator, D is the diffusion coefficient, S is the nutrient concentration in the 3D computational domain, and R is the consumption term. The substrate-specific

**Table 2. Growth parameters.**

| Species | Parameter | Symbol | Value | Unit | Reference |
|---------|-----------|--------|-------|------|-----------|
| *S. elongatus* | | | | | |
| | Maximal specific growth rate | $\mu_{max}$ | $1.89 \times 10^{-5}$ | $s^{-1}$ | [44], this work |
| | Biomass yield | Y | 0.55 | g-dw/g-co2 | |
| | Affinity constant for light | $K_{light}$ | $3.5 \times 10^{-4}$ | $kg\ m^{-3}$ | Fit from [46] |
| | Affinity constant for $CO_2$ | $K_{CO2}$ | $1.38 \times 10^{-4}$ | $kg\ m^{-3}$ | Fit from [46], S1 Fig |
| *E. coli* | | | | | |
| | Maximal specific growth rate | $\mu_{max}$ | $6.71 \times 10^{-5}$ | $s^{-1}$ | [36], this work |
| | Biomass yield | Y | 0.43 | g-dw/g-sucrose | [36] |
| | Affinity constant for $O_2$ | $K_{O2}$ | $1 \times 10^{-3}$ | $kg\ m^{-3}$ | This work |
| | Affinity constant for sucrose | $K_{sucrose}$ | 3.6 | $kg\ m^{-3}$ | This work |
| | Cellular maintenance | m | $9.5 \times 10^{-7}$ | $s^{-1}$ | This work |

consumption for cyanobacteria are as follows:

$$R_{light} = \left(-\frac{1}{Y}\right)(\mu_i + \psi)\rho \tag{5}$$

$$R_{CO_2} = \left(-\frac{1}{Y}\right)(\mu_i + \psi)\rho \tag{6}$$

$$R_{O_2} = \left(\frac{0.727}{Y}\right)(\mu_i + \psi)\rho \tag{7}$$

$$R_{sucrose} = \left(\frac{0.65}{Y}\right)\psi\rho \tag{8}$$

where $\rho$ is the density of cells within the grid. For all simulations, the computational grid was considered an optically-transparent, gas-permeable system (*e.g.*, a flask or microfluidic device). Light was a fixed concentration and was not a depletable resource (*i.e.* infinite source). This assumption was based on the thickness of the simulation volumes being 10 μm, within which we do not expect an appreciable attenuation in irradiance intensity [41,42]. In addition, due to limitations in the underlying LAMMPS framework used by NUFEB, light had no directionality, nor was a reduction in light intensity due to cell-cell shading possible [43]. The default concentrations and boundary conditions of all other nutrients can be found in Table 3.

The model for the heterotrophic *E. coli*, depends on sucrose and $O_2$:

$$\mu_i = \mu_{max}\left(\frac{[sucrose]}{K_{sucrose} + [sucrose]}\right)\left(\frac{[O_2]}{K_{O_2} + [O_2]}\right) \tag{9}$$

**Table 3. Default nutrient concentrations and boundary conditions.**

| Nutrient | Concentration (kg/m³) | Boundary Condition |
|----------|----------------------|--------------------|
| Light | $1 \times 10^{-1}$ | Dirichlet |
| $CO_2$ | $3 \times 10^{-2}$ | Dirichlet |
| $O_2$ | $9 \times 10^{-3}$ | Dirichlet |
| Sucrose | $1 \times 10^{-20}$ | Neumann |

$$R_{sucrose} = -\left(\frac{1}{Y}\right)\mu_i\rho \tag{10}$$

$$R_{O_2} = -0.399(\mu_i + m)\rho \tag{11}$$

$$R_{CO_2} = 0.2(\mu_i + m)\rho \tag{12}$$

where m is the cellular maintenance coefficient, $K_{sucrose}$ and $K_{O2}$ are the half-velocity constants for sucrose and $O_2$, respectively, which were determined from published data [36].

The simulation volume was 100 x 100 x 10 μm, unless otherwise specified, which was meant towards experimental co-cultivation and microscopy in similarly-sized microfluidic chambers. Additionally, this limited simulation volume served to minimize computational requirements. Simulations were run for up to 100 hours or $1.5 \times 10^7$ fg total biomass, whichever was sooner, to ensure the simulation volume did not become full.

## Model parameter optimization

In order to modify the cyanobacterial growth and sucrose secretion functions to reflect the relative induction strength of CscB/SPS, the model was fit to experimental data. To reduce the parameter-space, hyper-parameter optimization was performed using the Python package Scikit-optimize [47] in two steps, first on growth data, followed by sucrose secretion. The stochastic nature of initial cell biomass was accounted for by running a minimum of 3 simulations for each optimization step.

## Analysis of NUFEB simulations

As part of this work, the Python package nufeb-tools was developed to facilitate common computational operations, such as seeding new simulation conditions, ingesting the various NUFEB output files into Python, performing calculations, and generating visualizations [48]. The source code is available at https://github.com/Jsakkos/nufeb_tools.

## Spatial metrics for colony fitness

A variety of metrics were calculated to determine the impact of spatial structuring on colony fitness. Unless otherwise specified, all spatial metrics were calculated at the beginning of the simulation (t = 0 hours). The nearest neighbor distance was calculated by taking the minimum distance, d, between a colony and its neighbors. Colony-specific neighbor distances were calculated as well, denoted by the species the distances were calculated with respect to (e.g., Nearest neighbor s.e. indicates the distance to nearest *S. elongatus* was calculated and Nearest neighbor e.c. indicates the distance to nearest *E. coli* was calculated):

$$Nearest\ neighbor\ distance = \min (d_1, d_2, \ldots d_n) \tag{13}$$

The mean neighbor distance (i.e. inter-colony distance, $\bar{IC}$) was calculated as the sum of the distance between all colonies over the number of colonies [49]:

$$\bar{IC} = \frac{\sum_{i=1}^{n} d_1, d_2, \ldots d_n}{n} \tag{14}$$

The relative neighbor distance is defined as the distance to the nearest neighbor colony over the mean inter-colony distance:

$$Relative\ neighbor\ distance = \frac{\min\ (d_1, d_2, \ldots d_n)}{\bar{IC}} \qquad (15)$$

The inverse neighbor distance is defined as the inverse sum of the distances to all neighbor colonies:

$$Inverse\ neighbor\ distance = \sum_{i=1}^{n} \frac{1}{d_i} \qquad (16)$$

The log inverse squared neighbor distance is defined as the logarithm of the squared inverse sum of the distances to all neighbor colonies

$$Log\ inverse\ squared\ neighbor\ distance = log \sum_{i=1}^{n} \frac{1}{d_i^2} \qquad (17)$$

The scaled inter-colony distance accounts for the species-specific rate of primary nutrient diffusion (D) and the maximum growth rate ($\mu_{max}$) [49]:

$$\zeta = \frac{\bar{IC}}{\sqrt{\frac{D}{\mu\_max}}} \qquad (18)$$

Voronoi tessellation areas were calculated with SciPy [50].

## Colony fitness prediction

Simulations for predicting colony fitness were seeded randomly with 1–100 cells of each type within a 100 x 100 x 10 μm chamber and run for up to 100 hours or $1.5 \times 10^7$ fg total biomass, whichever was sooner. The data from 1,000 simulations was split into training (70%), validation (15%), and testing (15%) sets with the following number of samples: Train 69,204; Validation 14,829; Test 14,830. To perform the fitting, a sequential neural network was constructed with input normalization, 3 hidden layers consisting of 512 neurons each, with ReLU (Rectified Linear Unit) activation, layer dropout of 0.5, $L_2$ regularization of 1e-4, and batch normalization using TensorFlow [51]. Subsequent analysis of the prediction model's feature importance was performed using SHAP (SHapley Additive exPlanations) [52,53].

## Results & Discussion

### IbM and parameter fitting for *S. elongatus*

We sought to implement an IbM of the cyanobacterial co-cultivation platform to enable us to predict the impact of localized exchanges on species performance. For this purpose, we utilized NUFEB, an IbM framework capable of simulating physical, chemical, and biological processes [35]. NUFEB was developed to study emergent interactions by simulating cells in microbial consortia as individuals, each with the ability to sense and interact with their local environment, uptake and secrete nutrients as the result of metabolic processes, grow and divide, and even form biofilms. The light-driven platform we chose to model with NUFEB consists of a model cyanobacterium, *S. elongatus* CscB/SPS (see above), which performs oxygenic photosynthesis, fixes $CO_2$, and secretes sucrose [23], paired with a heterotrophic partner, *E. coli*, which can utilize sucrose as its sole source of carbon [36] and has previously shown to form a stable consortium [31]. We integrated custom models for each of these species into NUFEB

based on Monod kinetics, which relate the relative growth rate of microbes to the concentration of limiting nutrients [54].

To ensure that the IbM was biologically accurate, it was first necessary to determine key features related to the growth and metabolism of *S. elongatus*. The sucrose production and secretion pathway is inducible by the addition of IPTG, which drives the expression of SPS and CscB [25,26]. We therefore collected experimental data of sucrose secretion and cellular growth over time as a function of the IPTG concentration to evaluate the tunability of this platform (Fig 2A and 2B). The growth rates of differentially induced *S. elongatus* cultures varied minimally (Fig 2A, left), but we observed a slight growth decrease when the sucrose-secretion pathway was highly expressed (Fig 2A, right), consistent with prior results and indicative of the increasing metabolic burden of sucrose production. Sucrose accumulated upon induction, with up to 0.4 mM sucrose in the media after 24 hours (Fig 2B, left). We observed a logistic relationship between the induction level and extracellular sucrose after 24 hours and a 27-fold difference in the induction ratio (Fig 2B, right). Additionally, certain parameters were fit or derived from the literature (S1 Fig and Tables 1 and 2).

It has been previously demonstrated that engagement of this sucrose secretion pathway leads to a higher overall $CO_2$ fixation and improvement in total biomass accumulation [23,25,26,28]. By correlation with dry weight cell biomass, the impact of engagement of sucrose export on organismal fitness was determined. We observed an enrichment in the proportion of biomass directed to sucrose with increased IPTG induction (Fig 2C, left), which was in agreement with the simulated data (Fig 2C, right). An improvement in the total biomass accumulated after 24 hours of 17% was observed when comparing the maximum induction level (1 mM IPTG) to the uninduced control (0 mM IPTG), (Fig 2D). This phenomenon is likely due in part to increased overall photosynthetic flux and $CO_2$ fixation rates in *S. elongatus* when these pathways are engaged [25,26], is assisted by documented increases in Rubisco abundance following sucrose export [55] and potentially, that the specific ATP/NADPH requirements may alter the balance of metabolite pools important for bioproduction [56,57].

## IbM and Parameter fitting for *E. coli*

We likewise conducted a series of growth experiments for the heterotrophic partner species of our co-cultures, *E. coli*. We first wished to evaluate the capacity of *E. coli* to utilize sucrose in R2 medium. In agreement with previous studies [36,38], *E. coli* readily used sucrose for growth, with the relative growth rate dependent on the concentration of sucrose (Fig 3A). The cultures reached a maximum $OD_{600}$ of ~11. The sucrose concentration in the medium decreased proportionally to the culture densities, with an average level of ~1 mM after stationary phase was reached over time (Fig 3B). This data was used to fit the maximal growth rate ($\mu_{max}$), allowing for an approximation of growth rate as a function of local concentrations of soluble sugars. While the fit to the growth data was very good (Fig 3A, $R^2 = 0.98$), there was some disagreement between the simulated and experimental sucrose measurements (Fig 3B $R^2 = 0.84$), which was likely a result of additional sources of carbon in the medium (*e.g.*, yeast extract). Future optimization of the medium to maintain adequate sources of nitrogen and trace elements while minimizing sources of carbon is required to improve the model accuracy.

## *S. elongatus* sucrose secretion controls the ratio and dynamics of *E. coli* populations in co-culture

With the capacity to simulate growth dynamics of each species independently, *S. elongatus* and *E. coli* co-cultures can be modelled using a random initial positioning of individuals to extract unbiased simulation features and outcomes. Since the photosynthates produced by *S. elongatus*

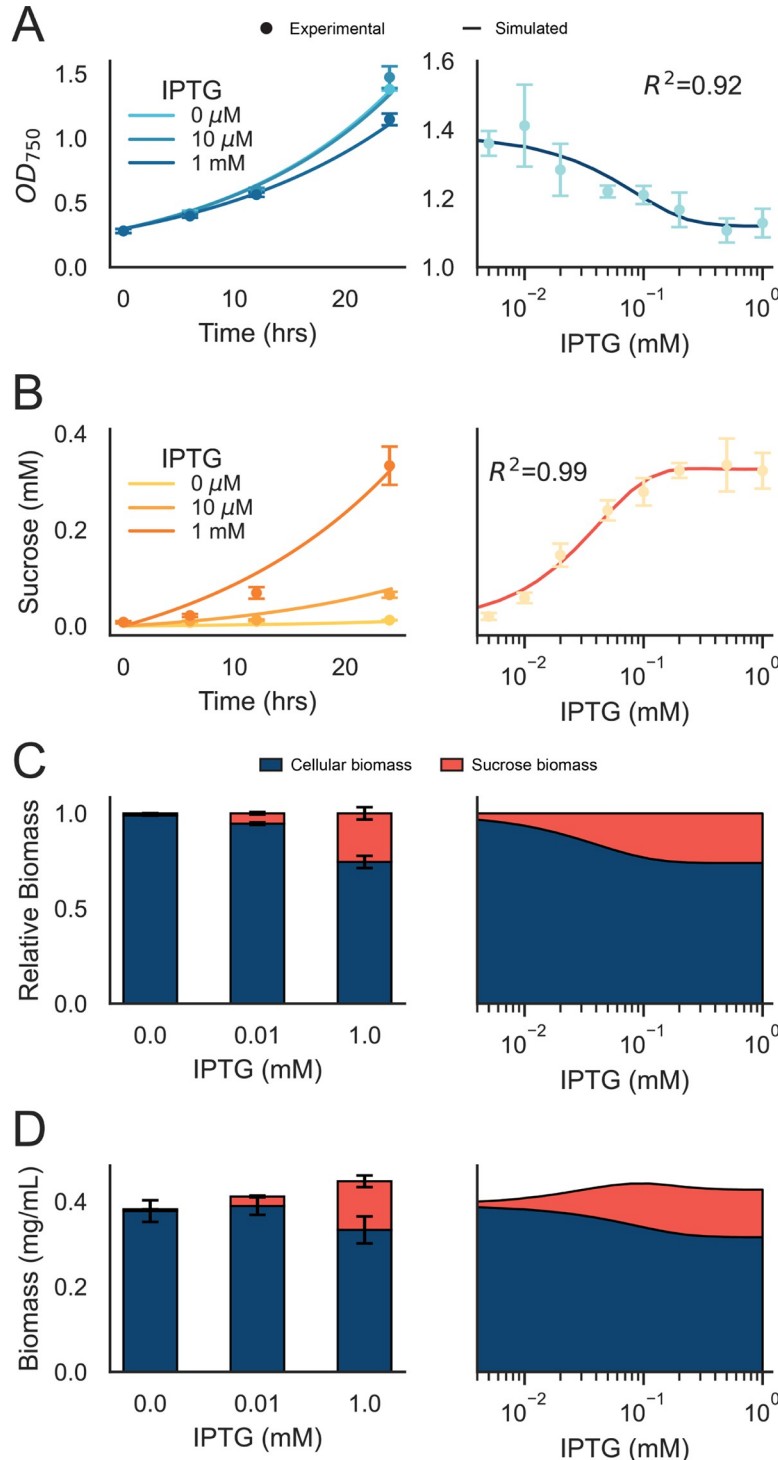

**Fig 2. NUFEB model validation with axenic *S. elongatus*.** A) Growth of *S. elongatus* over 24 hours under different levels of sucrose secretion induction (left). A small growth defect was observed after 24 hours of growth (right), indicative of the metabolic burden imposed by sucrose secretion. B) Sucrose secretion over time with varying induction (left). Extracellular sucrose after 24 hours of growth (right). C) Relative proportions of experimental (left) and simulated (right) cellular (blue) and sucrose (red) biomass. D) Total experimental (left) and simulated (right) cellular (blue) and sucrose (red) biomass. A/B) Dots indicate experimental data points (n>3), lines are the mean simulated result (n = 5), and the shaded regions are the standard deviation (n = 5). C/D) Shaded regions indicate the standard deviation (n = 5) and error bars indicate standard deviation (n = 3).

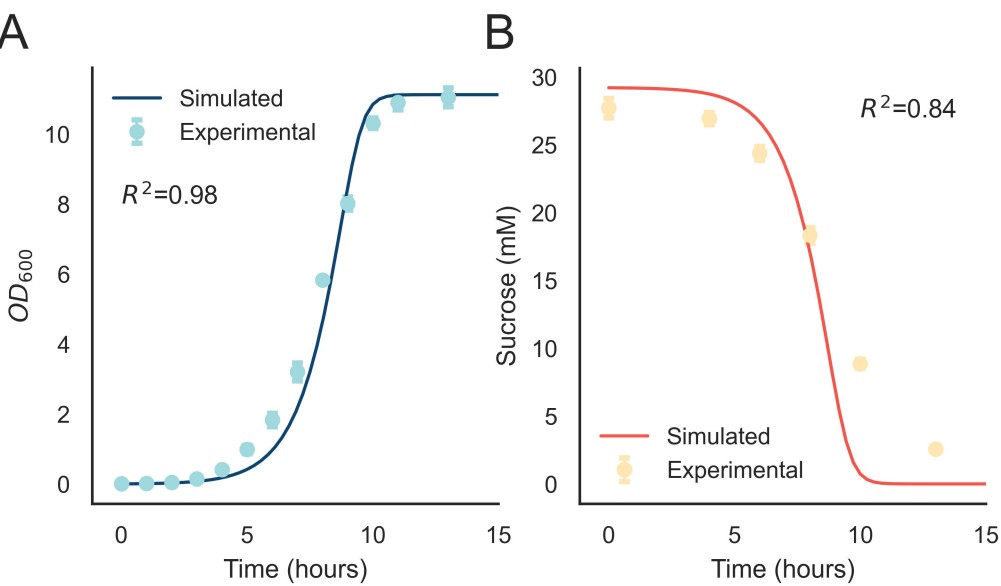

**Fig 3.** NUFEB model validation with axenic *E. coli* of both A) Growth (optical density at 600 nm) and B) Sucrose concentration over time. Error bars indicate standard deviation, n = 4. Shaded regions indicate standard deviation in simulated curves, n = 3.

represent the only available carbon source in the co-cultivation platform [23,36], the rate of *E. coli* growth should be constrained by the total photosynthetic activity. We simulated a series of co-cultivations under different induction regimes for sucrose export, while also varying the starting position and seeding ratio of *S. elongatus*:*E. coli.* As expected, the rate of sucrose export was strongly correlated with the rate of *E. coli* biomass accumulation (Fig 4A), while it was negatively correlated with *S. elongatus* cell biomass (Figs 2A, 2B and 4B). With a greater flux of sucrose, we expected increased accumulation of heterotrophic biomass, and the sucrose secretion level (IPTG) was positively correlated with the steady-state ratio of *E. coli*:*S. elongatus* biomass (Fig 4C and 4D). Despite considerable variations in the initial numbers of *S. elongatus* or *E. coli* cells, simulations routinely converged on a predictable ratio of *E. coli*:*S. elongatus* biomass within <3 days of simulated time (Fig 4D).

In addition to the changes in the simulations at steady state, we observed emergent behavior that was dependent on the level of sucrose secretion (Fig 4E–4G). Higher sucrose secretion led to transient accumulations of sucrose (Fig 4E), more rapid population stabilization (*i.e.*, average time required for population ratios to approach steady state; Fig 4F), with the average time required for the population to approach the steady state being inversely related to sucrose secretion rates, and an increase in the maximum sucrose concentration (Fig 4G). This behavior is indicative of the delay between when a public good becomes available and when partner microbes generate enough biomass to balance the flux of the public good.

To investigate the effect of the relative growth rates on the ability of *S. elongatus* to support heterotrophic biomass, we ran a series of simulations with varied $\mu_{max}$ for *E. coli* ($5 \times 10^{-2} < \frac{\mu_{max_{ec}}}{\mu_{max_{se}}} < 5 \times 10^1$). We observed a logistic relationship between the relative growth rates of the *E. coli* to *S. elongatus* (Fig 4H). Below a threshold, where the maximum growth rates were approximately equal, the population ratio dropped off precipitously. In this regime, *E. coli* cells did not grow rapidly enough to utilize the shared sucrose, and there was far less *E. coli* to *S. elongatus* biomass. In contrast, above the same threshold, the population ratio plateaued, indicating minimal additional

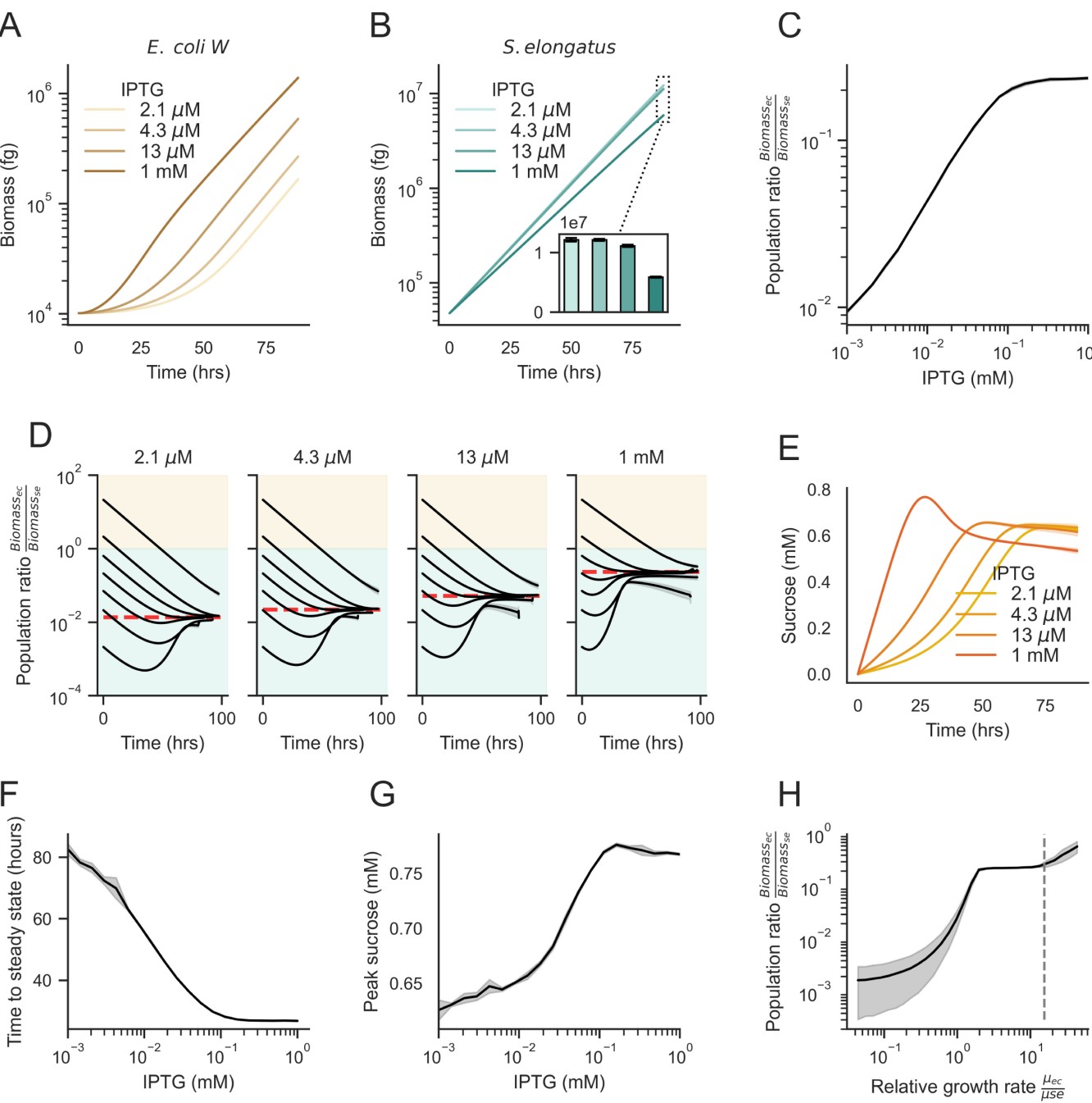

**Fig 4. Effect of sucrose secretion on simulated population-level dynamics.** Biomass over time with varying sucrose secretion for A) *E. coli* and B) *S. elongatus*. C) Steady state population ratio as a function of sucrose secretion. D) Population stability dynamics with varying sucrose secretion levels (IPTG, mM). Each curve represents a distinct starting ratio of *E. coli* to *S. elongatus* cells. Shaded regions indicate the dominant species (tan: *E. coli*, green: *S. elongatus*). E) Mean sucrose concentration over time. F) The effect of sucrose secretion on the time to reach steady state. Red shaded region indicates IPTG levels under which the simulated cultures did not reach steady state within 100 simulated hours. G) Peak sucrose concentration vs sucrose secretion level (IPTG, mM). H) The effect of relative growth rates ($\mu_{ec}/\mu_{se}$) on the steady state population ratio (1 mM IPTG). The dashed line indicates the relative growth rate used in all other simulations. In all panels except D, simulations were seeded with 50 cells of each type within a 100 x 100 x 10 μm chamber and run for up to 100 hours or $1.5 \times 10^7$ fg total biomass, whichever was sooner.

fitness benefits for heterotrophic partners to have division times that greatly outpace *S. elongatus*. This result suggests that when engineering new microbial communities, matching species such that the heterotrophs grow at least as quickly, and up to 10x faster, than their phototrophic partner will ensure maximal support for heterotrophic biomass.

## Effect of spatial proximity on colony-level fitness

Spatial structuring is thought to play a central role in promoting the stability of natural consortia, with diffusive and physical barriers strengthening local interactions. We sought to determine the impact of the initial spatial arrangement of the *E. coli* and *S. elongatus* consortium *in silico*. Towards this end, we ran 1,000 simulations (each up to 100 hours of growth), with randomly seeded initial cell numbers, size, locations, and IPTG concentrations. The spatial metrics (see Methods) were calculated for each colony and fed into a neural network-based regression algorithm (Fig 5A). Using the trained regression model, we can predict the aggregate colony fitness (total biomass at the end of the simulation) of both species in our simulations with R-squared values higher than 0.94, while species-specific predictions were slightly less accurate at 0.86 for *S. elongatus* and 0.80 for *E. coli* (Fig 5B). Cell species (Type) was the 2$^{nd}$ most important regression feature based on our SHAP analysis, which may explain the reduction in fitting accuracy for prediction species individually (S2 Fig), particularly given that it correlated with other features of importance (S3 Fig).

Initial biomass correlated with total colony biomass at the end of the simulation (fitness), albeit with a large variance (Fig 6A). *S. elongatus* and *E. coli* initial biomass value ranges were mutually exclusive, due to the differing sizes and densities of the respective species. IPTG had opposing sigmoidal relationships with *S. elongatus* and *E. coli* fitness (Fig 6B), as were expected based on the model design and experimental data. Increased sucrose secretion creates a small growth defect as a result of the metabolic burden (Fig 2A and 2B). In contrast, *E. coli* can utilize far more sucrose than *S. elongatus* cells were able to produce, and thus the shape of the *E. coli* fitness curve (Fig 6B, right) mimics the form of sucrose secretion with respect to IPTG (Fig 2B, right). $\bar{IC}_{se}$, the average inter-colony distance to *S. elongatus* colonies, was positively correlated with *S. elongatus* fitness, indicating that to some extent, phototrophic competition was detected (Fig 6C, left). We expected *S. elongatus* to be relatively indifferent to proximity of either species due to the lack of negative regulation in its computational model. Additionally, we observed a linear correlation between $\bar{IC}_{se}$ and *E. coli* fitness, which was indicative of increased fitness for *E. coli* colonies in closer proximity to *S. elongatus* neighbors (Fig 6C, right). We expect that at larger length scales, this effect would be increasingly prominent, as the diffusion gradients of sucrose, the limiting nutrient for *E. coli* growth, become steeper. $\bar{IC}_{ec}$, the average inter-colony distance to *E. coli* colonies, was not correlated with *S. elongatus* fitness, but was strongly correlated with *E. coli* fitness, due to the greater distance between colonies indicating less concentration of heterotrophic competitors (Fig 6D). Additional analysis of the feature and permutation importance from the regression model are shown in S2 Fig. Notably, Voronoi tessellation area, a geometric representation of the growth potential for a given colony, was an exceptionally poor predictor of fitness. Perhaps this was a result of having multiple species competing for space, yet participating in resource sharing, which would not be accounted for in a purely geometric evaluation of the microbial landscape. Finally, an illustrative example of the fitness differences between *E. coli* colonies is shown in S4 Fig.

## NUFEB model limitations and future directions

While the NUFEB framework itself is quite detailed, the underlying metabolic models used in this work were based on Monod kinetics, which only accounted for growth-limiting nutrients (i.e. light, $CO_2$, $O_2$, and sucrose). In addition, our simulations assumed saturated levels of

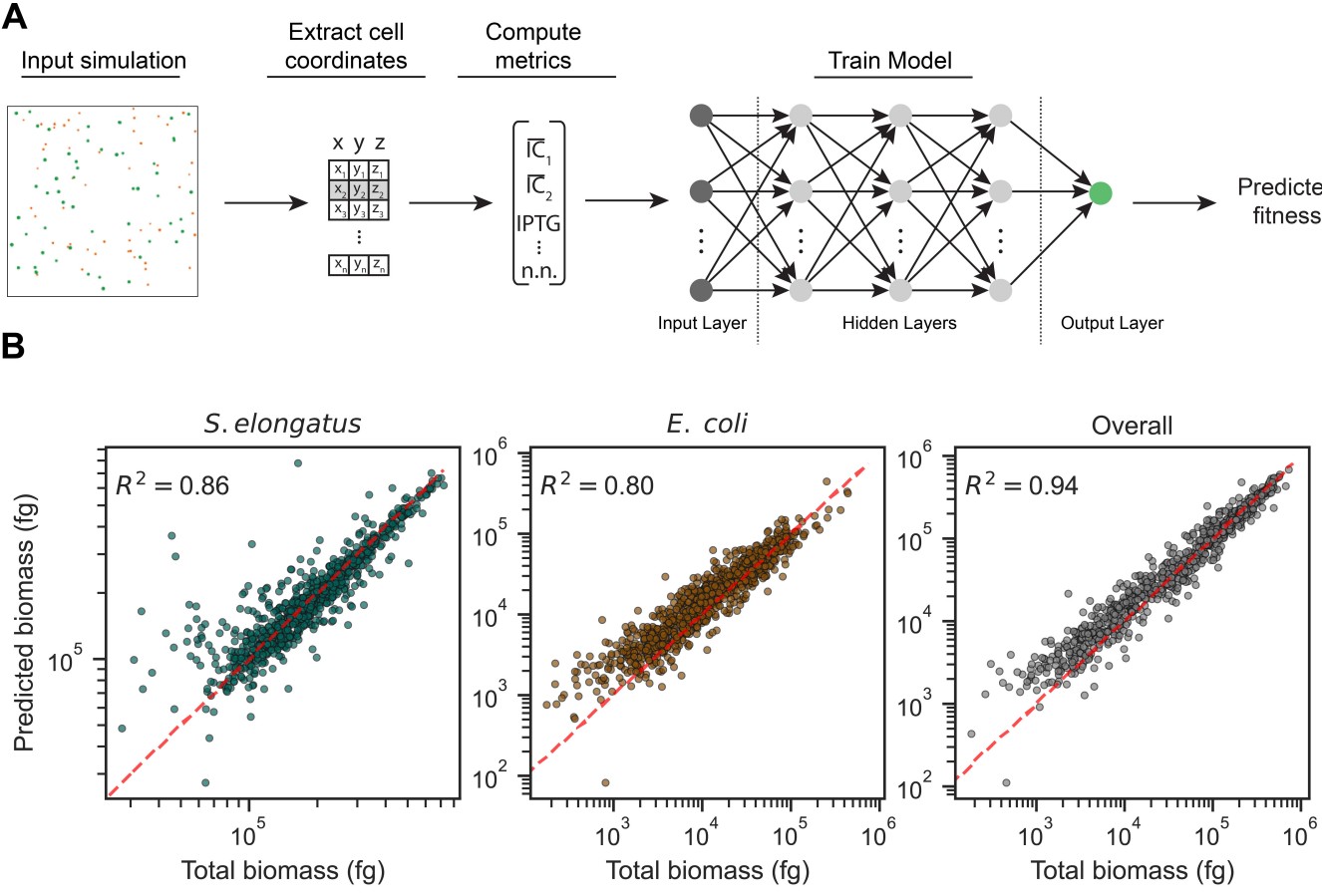

**Fig 5. Colony-level fitness prediction based on spatial metrics.** A) Diagram illustrating the prediction pipeline. Individual cell coordinates and sizes were extracted from the simulations, which were used to compute a variety of spatial metrics (see Methods section). A neural network regression model was subsequently fit to the aggregated spatial metrics, enabling fitness prediction. B) Actual vs. predicted biomass of *S. elongatus* (left), *E. coli* (center), and overall fit (right) in a simulated co-culture, with 1,000 datapoints shown. The data was split into training, validation, and testing groups as follows: Train 69,204; Validation 14,829; Test 14,830. Each dot indicates a sample colony. Dashed red line indicates a slope of 1. $R^2$ values indicate the coefficient of determination of the test set.

dissolved $CO_2$, which is likely to overestimate the potential cyanobacterial growth within a dense culture. Some disagreement between our experimental results and simulation data also comes from the difference in light availability. In flask experiments, self-shading causes a decrease in realized cell growth rates (Fig 2A), but due to the thin optical cross-section of the environments we examined herein, light shading by neighboring cells was neglected (see Materials and Methods) [41,42].

More complex metabolic models could be readily included within the NUFEB framework, including the possibility of incorporating entire genome-scale metabolic models within each individual [58]. Indeed, recent work from other laboratories utilizing this co-cultivation platform suggest that many other metabolites that are secreted from *S. elongatus* or co-cultivated heterotrophs could contribute to emergent interactions that positively influence species fitness [39]. Some potential emergent interactions that have been hypothesized include metabolite sharing, reactive oxygen species mitigation, and increased local concentration of $O_2/CO_2$ from photosynthesis/respiration [31,34]. Incorporating more detailed metabolic networks could further refine the predictive capacity of the simulations, but would be maximally useful if they are grounded in rigorous experimental datasets.

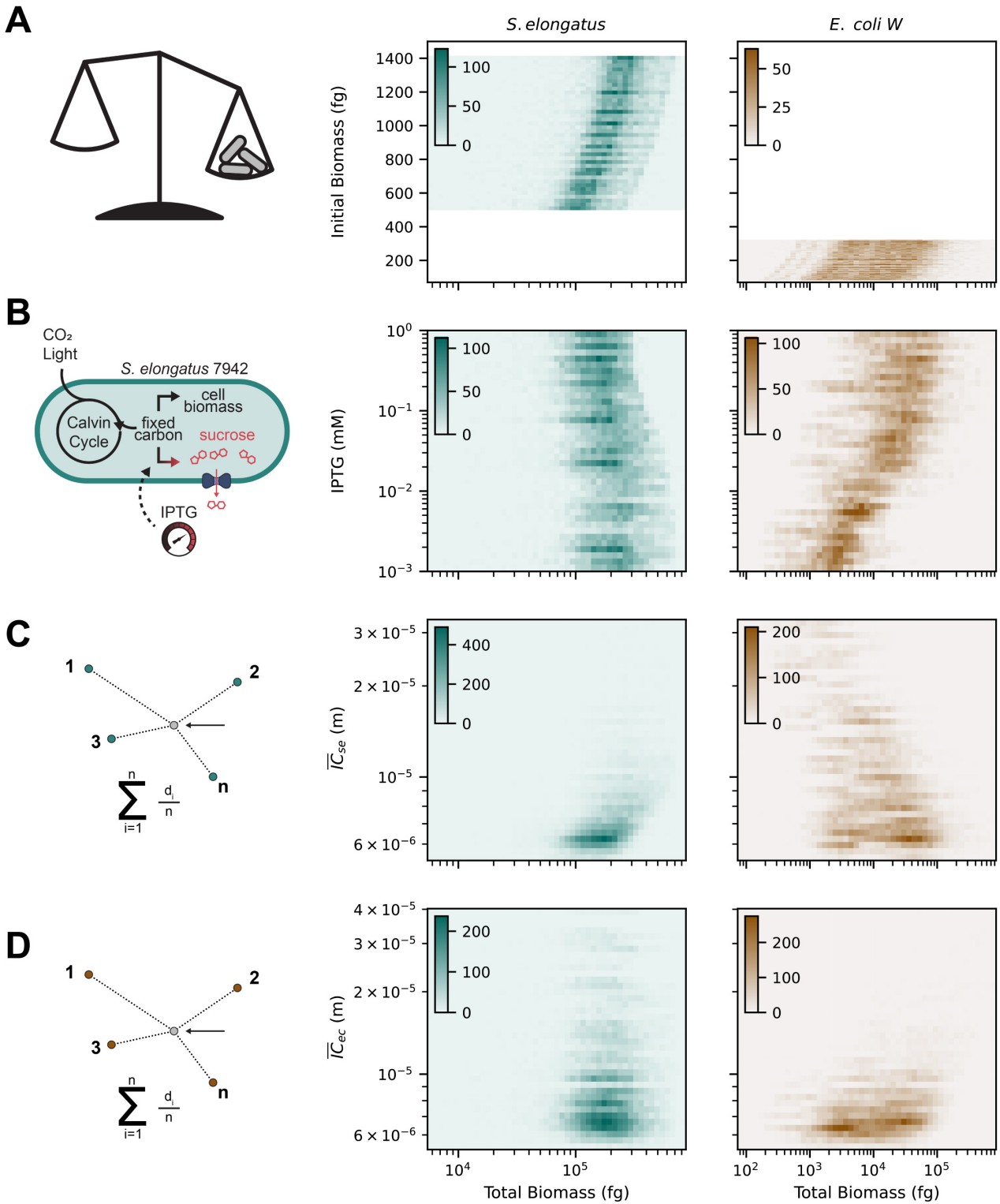

**Fig 6.** Two-dimensional histograms of the four significant parameters for determining colony fitness of *S. elongatus* (middle) *and E. coli* (right) colonies. The total biomass of each colony at the end of each simulation was used as a proxy for fitness. A) Initial biomass. B) IPTG concentration (sucrose secretion level). C) Mean inter-colony distance to S. elongatus. D) Mean inter-colony distance to *E. coli*.

In order for experimental co-cultures to be grown that have comparable growth kinetics to the simulations, a bi-species optimized medium is required. Our growth models were fit to experimental data on species-specific media optimized for axenic growth (*i.e.*, BG-11 and R/2). Previous studies have used modified BG-11, typically supplemented with additional sources of nitrogen [31,34]. While this does enable *E. coli* to grow, the density of their growth is more than an order of magnitude less than possible on rich media [36,38].

Finally, while spatial structuring plays an important role in partner fitness, using spherical representations of cells reduces our ability to observe and quantify more subtle effects, since cell morphology affects the resulting colony morphology [59]. A future NUFEB release will enable rod-shaped cells and allow more accurate spatial investigation.

## Conclusions

In this study, we developed an *in silico* model of a synthetic microbial consortium based on experimental data of sucrose-secreting cyanobacteria and *E. coli*. We showed that the level of sucrose secretion regulates the temporal dynamics of consortia growth, including the maximum sucrose concentration and time to steady state, and the amount of steady-state support for heterotrophic biomass. Based on spatial structuring, we fit a regression model to predict colony fitness. The critical parameters for fitness prediction were inter-colony distance, initial biomass, IPTG concentration, and distance from the center of the simulation volume. We expect that further integration of experimental data combined with computational approaches will improve our ability to design consortia with novel function.

## Supporting information

**S1 Fig. Fitting K$_{CO2}$ to experimental data [46], where** $\mu = \frac{[CO_2]}{K_{CO_2} + [CO_2]}$. **The best fit was K$_{CO2}$ =** $8.1 \times 10^{-3}$ g/L.
(EPS)

**S2 Fig. Regression feature analysis with SHAP (SHapley Additive exPlanations) values.** A) Feature importance. B) Permutation importance. This metric is a calculation in the drop in model accuracy after the data for the feature in question is randomly shuffled, and is complementary to feature importance. See materials and methods for definitions of each metric. s.e.–*S. elongatus*, e.c.–*E. coli*.
(EPS)

**S3 Fig. Pearson correlation matrix of the metrics used for fitness prediction.** The size and color correspond with the relative correlation magnitude. See materials and methods for definitions of each metric. S.e.–*S. elongatus*, e.c.–*E. coli*.
(EPS)

**S4 Fig. Comparing the relative fitness of winners and losers (*E. coli* W).** Colonies with higher initial biomass outcompete their neighbors. A) 2D representation of all colonies after 72 hours of simulated growth (1 mM IPTG). Cyanobacterial colonies are shown in teal and *E. coli* are shown in tan/brown. B) Growth of loser (i) and winner (ii) *E. coli* colonies over time, illustrating colony size and morphology. C) Biomass over time of the mother cells from the loser (i) and winner (ii) colonies (top) and total colony biomass over time. Dashed grey lines indicate a division event. D) Time between divisions for all mother cells, ranked from least to most fit.
(EPS)

## Acknowledgments

This work was supported in part through computational resources and services provided by the Institute for Cyber-Enabled Research (ICER) at MSU. Some of the simulations for this research conducted as part of a user project at the Center for Nanophase Materials Sciences (CNMS), which is a US Department of Energy, Office of Science User Facility at Oak Ridge National Laboratory. We thank Professor Joshua Vermaas for helpful discussions and assistance with NUFEB model optimization. We thank all members of the Ducat Lab for their input in editing the manuscript.

## Author Contributions

**Conceptualization:** Jonathan K. Sakkos, Bowen Li, Miguel Fuentes-Cabrera, Paolo Zuliani, Daniel C. Ducat.

**Data curation:** Jonathan K. Sakkos, Bowen Li.

**Formal analysis:** Jonathan K. Sakkos, María Santos-Merino, Emmanuel J. Kokarakis.

**Funding acquisition:** Daniel C. Ducat.

**Investigation:** Jonathan K. Sakkos, María Santos-Merino, Emmanuel J. Kokarakis, Paolo Zuliani.

**Methodology:** Jonathan K. Sakkos, María Santos-Merino, Emmanuel J. Kokarakis, Bowen Li.

**Project administration:** Miguel Fuentes-Cabrera, Paolo Zuliani, Daniel C. Ducat.

**Resources:** Daniel C. Ducat.

**Software:** Jonathan K. Sakkos, Bowen Li, Paolo Zuliani.

**Supervision:** Miguel Fuentes-Cabrera, Paolo Zuliani, Daniel C. Ducat.

**Validation:** Jonathan K. Sakkos, Bowen Li.

**Visualization:** Jonathan K. Sakkos.

**Writing – original draft:** Jonathan K. Sakkos, Daniel C. Ducat.

**Writing – review & editing:** Jonathan K. Sakkos, Daniel C. Ducat.

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
