## [Decision Letter · Decision Letter 0]

13 Dec 2022

Dear Dr. Ducat,

Thank you very much for submitting your manuscript "Predicting partner fitness based on spatial structuring in a light-driven microbial community" for consideration at PLOS Computational Biology. As with all papers reviewed by the journal, your manuscript was reviewed by members of the editorial board and by several independent reviewers. The reviewers appreciated the attention to an important topic. Based on the reviews, we are likely to accept this manuscript for publication, providing that you modify the manuscript according to the review recommendations.

Sincerely,

Pedro Mendes, PhD

Academic Editor

PLOS Computational Biology

Jason Haugh

Section Editor

PLOS Computational Biology

Reviewer's Responses to Questions

**Comments to the Authors:**

Reviewer #1: I liked how the model was calibrated using experimental data. There was a good synergy between experimental and computational techniques. I also appreciate the python code for processing of the NUFEB results.

Minor comments below:

Line 171: Is it mu or mu_i?

Lines 186-189: For equations 5-8, psi is used instead of psi_i, is there a difference? If yes, what is the formula for psi?

Line 282: Are there links to the custom models mentioned? Or sample NUFEB input files?

Line 305: Did you mean sucrose biomass accumulated?

Line 339: It is unclear how fig. 4B shows the negative correlation.

Reviewer #2: In the manuscript “Predicting partner fitness based on spatial structuring in a light-driven microbial community, the authors posed a computational IBM using a published platform NuFeb and parametrized with experimental data of Synechococcus elongatus a sucrose-secreting strain and Escherichia coli W that can use sucrose as a carbon source. Authors find that initial biomass and distance from the E coli colonies are the main parameters to establish successful consortia.

This is a good example of how simulations, particularly IbM models, can help us understand and predict how synthetic microbial consortia can be established; it is interesting how the author adapts the IbM model to explore these consortia, and I think it could be of interest of the academic community.

Some comments.

1.- Parameters were obtained from experimental data but in different culture conditions (different media). This is problematic for the parameterization and the prediction of establishing a stable consortium. If these two strains have already proven to form a consortium, why not grow them in the same media culture?

2.- The authors talk about the stability of natural consortia. As the simulations run to a steady state. It would have been interesting to subject the simulations to a change of environment, to challenge this stability. Can this be done on this platform?

3.- For measurements and validation of the IBM model, how many cells are in each experiment? 11 OD is a very high measurement; this could easily measure with a flow cytometer and correlated with the total number of cells in the computational simulation. These results could be relevant as the prediction from the model is that initial density is a determinant in the establishment of the consortia and also the accumulation of sucrose is also an important factor.

4.- If E coli utilize far more sucrose than S. elongatus can produce, do the authors obtain conditions by which there is competitive exclusion?

5.- I think the discussion could be improved by discussing how to integrate the findings in real experiments, as it has already been published that these two strains can be co-culture.

6.- A flask with constant shaking is usually used as a proximation of a homogeneous environment; why is it a spatial environment? Does any of these strains form spatially structured formations as biofilms? Consortia can survive in a homogenous environment, or if sucrose can diffuse further?

7.- In terms of total biomass (both strains), what conditions maximize it?

There is a trade-off between the secretion of sucrose ( high concentration of IPTG) and the growth of S. elongatus, so is there an optimal secretion rate?

Reviewer #3: The authors put together a novel experimental and computational analysis of a spatial microbial community. This is a very interesting paper and I think will be of interest to the Plos Comp bio community.

The authors used experiments to calibrate their individual based model and determined the importance of spatial organization towards predicting colony fitness.

Major Questions:

(1) One major question the author's sought to investigate how well fitness could be predicted. While I believe the authors have explained many of their technical details, I found it difficult - with the ordering of sections in their manuscript - to understand what they meant by "fitness"?

This is because it wasn't until page 20 (line 375) that the authors state "aggregate colony fitness (total biomass at the end of the simulation".

(2) I found it very interesting that the Voronoi tessellation area (aka the growth potential) for a given colony was a poor predictor of fitness (which I'm now interpreting as total colony biomass). Presumably this would not be the case for a different type of microbial species interaction?

(3) While the authors did an excellent job of using their framework in a specific context, it would be helpful for the readers if the authors could speak a bit more broadly about how this work might generalize to other types of microbial interactions? Or indeed would all future study require the same detailed fine-tuning the authors performed here?

Minor Edits:

- When equations end sentences, they should have a period after them. Periods are missing on Equations (12)-(17)

**Have the authors made all data and (if applicable) computational code underlying the findings in their manuscript fully available?**

Reviewer #1: **No: **Sample input files of the custom models haven't been provided.

Reviewer #2: Yes

Reviewer #3: Yes

PLOS authors have the option to publish the peer review history of their article (what does this mean?). If published, this will include your full peer review and any attached files.

Reviewer #1: No

Reviewer #2: No

Reviewer #3: No

Figure Files:

Data Requirements:

Reproducibility:

References:

---

## [Editor Report · Decision Letter 1]

22 Mar 2023

Dear Dr. Ducat,

We are pleased to inform you that your manuscript 'Predicting partner fitness based on spatial structuring in a light-driven microbial community' has been provisionally accepted for publication in PLOS Computational Biology.

Best regards,

Pedro Mendes, PhD

Academic Editor

PLOS Computational Biology

Jason Haugh

Section Editor

PLOS Computational Biology

---

## [Editor Report · Acceptance letter]

28 Apr 2023

PCOMPBIOL-D-22-01476R1 

Predicting partner fitness based on spatial structuring in a light-driven microbial community

Dear Dr Ducat,

I am pleased to inform you that your manuscript has been formally accepted for publication in PLOS Computational Biology. Your manuscript is now with our production department and you will be notified of the publication date in due course.

With kind regards,

Timea Kemeri-Szekernyes
